# A Statistical Analysis of Response and Recovery Times: The Case of Ethanol Chemiresistors Based on Pure SnO_2_

**DOI:** 10.3390/s22176346

**Published:** 2022-08-23

**Authors:** Andrea Ponzoni

**Affiliations:** 1National Institute of Optics (INO) Unit of Brescia, National Research Council (CNR), 25123 Brescia, Italy; andrea.ponzoni@ino.cnr.it; Tel.: +39-030-3711440; 2National Institute of Optics (INO) Unit of Lecco, National Research Council (CNR), 23900 Lecco, Italy

**Keywords:** metal oxides, chemiresistors, response time, recovery time, nanowires, nanoparticles, ethanol, diffusion, thermo-diffusion

## Abstract

Response and recovery times are among the most important parameters for gas sensors. Their optimization has been pursued through several strategies, including the control over the morphology of the sensitive material. The effectiveness of these approaches is typically proven by comparing different sensors studied in the same paper under the same conditions. Additionally, tables comparing the results of the considered paper with those available in the literature are often reported. This is fundamental to frame the results of individual papers in a more general context; nonetheless, it suffers from the many differences occurring at the experimental level between different research groups. To face this issue, in the present paper, we adopt a statistical approach to analyze the response and recovery times reported in the literature for chemiresistors based on pure SnO_2_ for ethanol detection, which was chosen as a case study owing to its available statistic. The adopted experimental setup (of the static or dynamic type) emerges as the most important parameter. Once the statistic is split into these categories, morphological and sensor-layout effects also emerge. The observed results are discussed in terms of different diffusion phenomena whose balance depends on the testing conditions adopted in different papers.

## 1. Introduction

Response and recovery times (t_RES_ and t_REC_) are among the most important parameters for gas sensors. The former accounts for the capability of the device to promptly alert about the presence of the target compound, and it has a strong relevance in determining the capability to promptly react and/or adopt countermeasures [1,2,3]. The latter accounts for the system readiness in repeated measurements and is a fundamental parameter to determine the sensing-system throughput, which is particularly important in screening analyses such as those carried out in the medical and agrifood fields [4,5,6]. 

An important field of research to improve these functional parameters with respect to their state of the art is related to the development of materials with well-controlled morphological, compositional and structural features [7,8]. Focusing on metal-oxide (MOX)-based gas-sensors, features such as the oxide morphology, grain size, porosity and pore distribution and the addition of dopants or surface catalysts are widely investigated to optimize the functional properties of the gas sensor device [9,10,11]. This goal is typically pursued in individual papers comparing the effects induced by one or some of these features. In addition, tables comparing the recorded performance with those reported in the literature are often used to frame the obtained results with those reported by other research groups. Several papers indicate the morphology of elementary nanostructures as a key factor to achieve the desired functionalities, including the reduction of response and recovery times [12,13]. Indeed, intense research has been carried out to compare the performance of nanostructures such as nanoparticles, nanowires and nanosheets [14,15].

The comparison between the results achieved by different research groups is both fundamental and difficult. It is a mandatory task in order to attempt to approach a unitary vision between the different approaches and solutions reported in the literature. At the same time, the different measurement setups and device layouts used by different groups are very likely to affect the performance of the investigated materials and devices. Their effects may thus overlap with those arising from the targeted material properties. Looking at the aforementioned tables, the impression is that these experimental differences are often underestimated. In some cases, important information is not available, even from the original reference paper. In order to reduce the effects of such unknown differences as much as possible and obtain reliable conclusions, statistics are the most effective tool. 

In this work, we use a statistical approach to screen and compare the effects arising from parameters such as the morphology of elementary nanostructures, the gas concentration, the working temperature, the measurement setup and the device layout on the response and recovery times. The analysis is focused on chemiresistor devices based on pure SnO_2_ tested against ethanol, which is chosen as a case study. SnO_2_ was chosen because it is the most studied material among metal oxides and offers a suitable statistic [16,17]. Similarly, ethanol was considered owing to its frequent use as a test molecule in gas sensing and to the large amount of data available in the literature. 

The rest of the manuscript is organized as follows: Section 2 describes the different types of mechanisms reported in the literature to affect the response and recovery times of metal oxide chemiresistors; Section 3 describes the statistical methods used to carry out the analysis; the results are reported in Section 4 and are discussed in terms of models and literature findings in Section 5; the conclusions are in Section 6. 

## 2. Theoretical Background

The experimental values recorded for the response and recovery times of gas sensors are always given by the convolution between the intrinsic time constants of the device and those of the measurement setup. Each term may be further decomposed into contributions from more elementary processes that depend on the specific features of the considered setup, device, material and gas–MOX interaction.

In order to provide the background for discussing the results presented in the next sections, this paragraph reports a brief resume of the main mechanisms underlying the transients of response and the recovery phenomena.

### 2.1. Flow and/or Diffusion inside the Test Chamber 

The target gas is injected into the test chamber (or the background atmosphere is restored inside it) from a single point-like injector/extractor, and a certain time is required before the atmospheric composition reaches its final steady state composition inside the whole volume of the chamber. To determine the temporal and spatial evolution of the atmospheric composition, flow equations should be combined with the diffusion-convection laws [18]. The former explains the motion of the gas molecules in the chamber carried by the jet flow [18,19]. The latter explains the gas flux $\vec{J}$ arising from the diffusion phenomena stimulated by the gradient of the gas concentration C: $\vec{J}=-D\vec{\nabla }C$, where D is the diffusion coefficient of the target gas inside the background fluid (air, in the present case). D is often expressed by the Einstein–Stokes equation, $D=k_{B}T_{E}/\left(3\pi \eta d\right)$, where k_B_ is the Boltzmann constant, T_E_ is the environmental temperature in Kelvin degrees, η is the viscosity of the air background and d is the diameter of the target gas molecule [20]. The results reported in the literature show that the distribution of the gas concentration inside the chamber volume and its time dependence have a strong dependence on the shape, size and fluid dynamics of the measurement conditions and setup [19,21,22,23]. The diffusion or flow through the pipelines connecting the chamber to the source of gases (certified bottles or syringe inlets) also play a role. For small and properly designed test chambers, gas molecules directly reach the sensor through a laminar flow with minimal recirculation effects and diffusion [18,19]. In test chambers not specifically optimized for this purpose, such as those typically used in the literature, setup transients may reach the order of hundreds of seconds. This has been proven by both simulations and experiments [18,19]. Nonetheless, it should be mentioned that, while simulations may easily focus on the target flow/diffusion phenomena, in experiments, these effects are much harder to decouple for the other effects discussed in the following part of Section 2. 

Schematic representations of gas diffusion and gas flow inside the test chambers are shown in Figure 1, considering, in both cases, certified bottles as gas/air sources. 

### 2.2. Diffusion Phenomena in Proximity of the Metal Oxide Layer

In setups exploiting sensors with an embedded heater or with a small heater hosting the sensor device, a temperature gradient is established between the sensor and the surrounding atmosphere, as schematically reported in Figure 2a. This gradient impacts the diffusion of gaseous molecules from and to the sensor surface in a complex manner, coupling with the flow and diffusion-convection equations described in Section 2.1. Detailed formulations of the differential equations describing these phenomena can be found in the dedicated literature [24,25]. The results show the tendency of molecules to move either from or to the warmer region, depending on properties such as the global fluid compressibility and the molecular weight of the considered chemical species [24]. This is quite interesting for gas sensing since it introduces an asymmetry between the response and recovery times. In detail, this effect will promote faster response times and longer recovery times for target molecules tending to approach the warmer region and vice versa for target molecules showing the opposite tendency. 

Despite the general solution of these phenomena requiring numerical simulations, some papers report specific experiments designed in an ad hoc manner to reduce all flow/diffusion phenomena other than those stimulated by the temperature gradient as much as possible [20,26]. In these cases, the gas flow is described by a simple equation, $\vec{J}≅-D_{T}\vec{\nabla }T$, where $D_{T}=D/T$ is the thermo-diffusion coefficient and T is the local temperature [20,26]. Given the similarity of this equation with the diffusion equation, this phenomenon is often named thermo-diffusion. The values reported in the literature for thermo-diffusion, D = T × D_T_, are of the order 10^−4^–10^−8^ m^2^/s [20,27,28]. The size of the region around the warm sensor, in which there is the temperature gradient (L_ΔT_), can be estimated as being of the order of a few mm, which correspond to an estimated time $t_{\Delta T}≅{\;L}_{\Delta T}^{2}/D$ of the order of 0.01–100 s. It should be kept in mind that the experiments and simulations dedicated to the thermo-diffusion effects are typically carried out in conditions quite different from those adopted in gas sensing; hence, the estimated timescale should be considered with prudence. At the very least, these values do not permit the a priori exclusion of thermo-diffusion from the phenomena affecting the transients of gas sensors. 

### 2.3. Diffusion Phenomena through the Pores of the Metal Oxide Layer

Most of metal oxide chemiresistors are based on metal oxide layers featuring a certain degree of porosity. Porosity and pore size distribution are fundamental parameters to ensure that, once gaseous molecules have approached the sensor device through the phenomena discussed in the previous Section 2.1, they will also easily reach all of the nanostructures that compose the sensing layer. Diffusion inside pores is typically described as stimulated by the gas concentration gradient, following the same Fick’s equation reported in Section 2.1 but with a different diffusion coefficient owing to the different regime(s) of diffusion. Indeed, as the size of the medium in which diffusion takes place (in this case, the pore radius, r_p_) approaches the mean free path of the molecules, L_mfp_, the diffusion enters the molecular and the Knudsen regimes. Indicatively, these regimes are typically quoted to apply for L_mfp_/2r_p_ ≤ 0.1 (molecular) and for 0.1 < L_mfp_/2r_p_ < 1 (Knudsen), [29]. Lmfp=kBT2πpd2, where p is the pressure of the considered environment and the other symbols have the same meaning as in the previous sections. At the typical working conditions of MOX chemiresistors (p ≅ 1 atm, T_S_ in the range 200–500 °C), referring to ethanol, which has d ≅ 0.45 nm [30], the L_mfp_ lies in the range 70–120 nm. Considering the typical pore size of metal oxides [17], the diffusion is of the Knudsen type, and the related diffusion coefficient at the level of individual pores is DK=2rp38RTπM, where R is the gas constant and M the molecular mass of the diffusing species. However, in sensitive layers featuring a very open morphology, such as those often obtained with nanowire and nanosheet networks, intersecting crystallites leave large voids that may largely exceed the 100 nm size. In these cases, diffusion is of the molecular type, and molecular collisions are more likely to occur than collisions with pore walls. The diffusion coefficient becomes DM=Lmfp38RTπM. 

Both types of diffusion are favored by increasing the sensor temperature, with D_K_ and D_M_ increasing according to T^1/2^ and T^3/2^, respectively.

Relating the diffusivity values to the experimental data requires the estimation of the effective diffusivity, which should account for the disordered network of interconnected pores. From a modelling point of view this is a complex task and has been the subject of dedicated studies in the specialized literature. These works employed ad hoc-designed experimental conditions that are different from those encountered in gas sensing [31]. From an experimental point of view, several gas-sensing papers reported on the importance of pore size and distribution in improving the performance of metal oxides, showing beneficial effects in terms of both sensor response intensity and response/recovery speed [32,33]. In some cases, multimodal pore distribution was specifically indicated as the main feature underlying the experimentally observed improvements owing to its capability to accommodate the requirements of diffusion through the sensing layer at different length scales [34,35]. The measured response and recovery times interpreted based on diffusion through the porous layer vary in a broad range. Some papers report values of the order of a few or a few tens of seconds [33,36], while others report the response time decreasing from a few tens of seconds to a few seconds and the recovery times decreasing from a few hours to a few minutes depending on the layer porosity [34]. These variations may reasonably depend on the variety of the pore network found in different layers as well as on the coupling with other phenomena. 

### 2.4. Interactions between the Gas Molecules and the Metal Oxide

Gas molecules undergo flow/diffusion phenomena, as described in the previous sections, approach the surface of the metal oxide and interact with it, giving rise to the sensor response. This interaction occurs through the combination of several phenomena of a chemical and electrical nature [16,37]. The response to a reducing gas, e.g., ethanol, involves the chemical adsorption of target molecules, their oxidation through the interaction with chemisorbed oxygen species available over the oxide surface and the release of electrons from surface states into the conduction band of the semiconductor. Intermediate species may also be developed during this process [37,38]. The sensor recovery upon the removal of the target gas from the test chamber involves the desorption of eventual by-products and the chemisorption of oxygen from the gas phase to restore the population of oxygen ions that was perturbed by the target gas. These phenomena involve the creation of surface states related to chemisorbed oxygen and the trapping of electrons from the semiconductor conduction band. Despite the complexity of these interactions, they can be conveniently expressed through effective adsorption and desorption rates, k_ADS_ and k_DES_, that account for the whole ensemble of chemical and electronic phenomena. Both rates are thermally activated and follow an Arrhenius-type temperature-dependence, kx=Ax exp(−Eatt,xkBTS), where the subfix ‘x’ stands for ADS or DES, depending on the considered process, A_x_ is a constant and E_att,x_ is the effective activation energy [39,40,41]. 

To bypass the flow/diffusion phenomena discussed in Section 2.1 and obtain response and recovery times depending only on the interaction kinetics and diffusion through and within the sensing film, dedicated setups have been developed. These exploit either a very small test chamber (≤mL) with minimized dead volumes/pipelines and large flows (>>mL/s) [34] or an open environment with gas injectors flowing the gas jet (or the background-atmosphere jet for the recovery process) directly over the sensor surface [42,43]. In some cases, to further reduce the diffusion processes occurring inside the sensing layer (Section 2.2), thin films were employed instead of porous ones [43]. Using these measurement methods, response and recovery times down to 0.1 s were recorded [43], with the transient becoming slower with decreases in the sensor temperature and reaching values of the order of 10^3^ owing to the strong temperature dependence. 

It is further worth noting that the effective activation energy of these interactions is typically larger for the desorption process than it is for the adsorption process: E_att,DES_ > E_att,ADS_, which means k_DES_ < k_ADS_ and, in turn, t_REC_ > t_RES_ [44]. This introduces an asymmetry between the response and recovery times. 

Interestingly, the interaction rates can be directly coupled with diffusion through the MOX layer in a single diffusion type equation characterized by an effective diffusion coefficient $D_{E}\approx D_{K}/\left(1+k_{\mathrm{ADS}}\left[S\right]/k_{\mathrm{DES}}\right)$. In this latter equation, the Knudsen diffusion inside pores is arbitrarily assumed (but it may be equally written for the molecular regime using the D_M_ coefficient instead of D_K_), and [S] indicates the concentration of the adsorbing sites over the MOX surface [36]. The coupling between diffusion and reaction rates becomes more effective as the pores get smaller. For pore sizes of the order r_p_ < 2 nm, the diffusion becomes of the surface type [29], which, in contrast to the other diffusion types discussed in Section 2.3, features an Arrhenius temperature dependence. 

### 2.5. Diffusion Phenomena through the Bulk of Individual Nanostructures

The interactions between the metal oxide and the gas molecules are usually described through the surface reactions discussed in Section 2.4. However, a deeper investigation shows that the oxygen diffusion through the bulk of individual nanostructures should be considered to reach the equilibrium between the surface and the bulk of the metal oxide [45]. This is both for the sensing oxygen itself and for gases other than oxygen. Indeed, compounds such as ethanol reduce the oxide surface by decreasing the population of oxygen ions, and the oxygen should be re-equilibrated as well. 

The diffusion of oxygen ions through the solid is a thermally activated process, which feature an Arrhenius-like behavior with an activation energy of about 0.9 eV [46]. The diffusion coefficient D_ox_ is around 10^−15^ cm^2^/s at room temperature and increases to D_ox_ ≅ 10^−13^ cm^2^/s at 300 °C [47]. The characteristic time associated with the transient to the equilibrium is of the order of $t_{\mathrm{ox}}≅{\;d}_{c}^{2}/(4{\times D}_{\mathrm{ox}})$, where d_c_ is the diameter of the individual nanostructure. Considering that d_c_ usually lies in the 10–100 nm range [17], at the sensor temperature of 300 °C, the order of t_ox_ falls in the 1–100 s range, with t_ox_ increasing with the square of d_c_. 

## 3. Materials and Methods

As detailed in Section 2, several phenomena contribute to the measured response and recovery times of gas sensors. In some cases, the exponential rise/decay functions have been reported to provide a satisfactory fit for the measured transients [39,41,48]; however, owing to the complexity of the involved phenomena, these simple functions are not always satisfactory, and more complex functions or even numerical solutions should be adopted to provide a satisfactory fit [19,36,39]. The challenge of modelling and fitting the sensor kinetics is probably one of the major motivations that led to the common practice of reporting response and recovery times through empirical parameters. It has become a consolidated practice to calculate these parameters as the time required by the sensor response to reach 90% of the variation between the initial and final steady states.

The experimental data analyzed in this paper are all of this kind. The full list of articles is reported in the Appendix A. The core of this list consists of papers indexed in ISI Web of Science as ‘articles’ published between 2015 and 2020 about chemiresistors based on pure SnO_2_ materials tested against ethanol in an air background. If necessary, response and recovery times were extrapolated from figures using the Engauge digitizer software [49].

Several parameters are considered to investigate those dependencies that emerge from the analyzed literature. These include both numerical and categorical variables, as detailed below.

Numerical variables:
Ethanol concentration, C_EtOH_, expressed in parts per million (ppm);Sensor temperature, T_S_, which may be different from the environmental temperature (T_E_) depending on whether the sensor temperature is controlled through a heater embedded in the sensor device or through a furnace;Pore radius, r_p_, which is the peak value typically extrapolated from N_2_ adsorption/desorption measurements;Crystallite size, d_c_, which is the smallest size of elementary crystallites. It is the diameter for nanoparticles and nanowires, while it is the thickness for nanosheets;Chamber volume, which is the volume of the chamber where sensors are lodged for gas sensing tests;Flow, which is the flow used to supply the target gas and to restore the baseline (for those setups employ a gas flow; see the ‘measurement method’ variable for details).


Categorical variables:
Morphology of elementary crystallites composing the sensing layer, which we divided into three classes, namely:
○nanoparticles, i.e., crystallites with a rounded shape [50,51,52];○nanowires, i.e., crystallites featuring an elongated shape with a length largely exceeding the diameter [53,54,55];○nanosheets, i.e., crystallites featuring a thin and large shape, with the thickness much lower than the other two dimensions [56,57,58].
Measurement method, which is roughly classified into two major classes:
○dynamic, often named the ‘flow through method’, refers to those setups employing a constant flow of gas through the test chamber [50,59,60]. Mass flow controllers are used to mix fluxes from certified bottles and control the atmosphere composition inside the test chamber. In these setups, the atmosphere surrounding the devices is continuously renewed by the injected flow, both when the atmosphere composition is changed as well as when the composition is kept constant. During the gas injection process, the gas concentration inside the flow is kept constant at the desired value. The device is immediately exposed to the target concentration if it is under a direct flow; otherwise, if the chamber is designed to involve the diffusion process, these should take place before the desired concentration is established in proximity of the sensor device;○static, which refers to those setups in which the target gas is injected inside the test chamber through a device, such as a syringe [61,62], an evaporating system pre-filled with a proper amount of liquid [63] or a certified bottle [64], which is actuated only at the time of gas injection. After the quick injection, which causes the gas concentration at the time and place of the injection to be much larger than the equilibrium value, the atmosphere is allowed to reach the final homogeneous composition in the whole volume by diffusion. Concerning gas removal, the baseline atmosphere is quickly changed inside the chamber and allowed to reach the steady-state, homogeneous distribution with no flow.
Heating method, which is classified in the following three classes:
○furnace, for those sensors that do not exploit a local heater but are lodged inside a furnace that warms the whole test chamber [65,66,67,68];○holder, for those sensors heated through a local gauge integrated in the measurement setup, particularly in the sensor holder [32,69,70,71];○meander, for devices realized using a flat ceramic substrate over which the heater, typically with the shape of a metallic meander, is deposited on one of the two faces [50,61,72];○coil, for devices realized using a tubular ceramic substrate, in which a coil-shaped heater is inserted [10,52,73,74].


For all the categorical parameters, the class ‘na’, standing for ‘not available’, has been used for those papers in which the considered information was not reported.

In some cases, the comparison between variables has been carried out based on the Mood’s median test—in particular, the mediantest.m function of MATLAB was used [75]. As will be discussed in Section 4, this test has been preferred over other, more powerful tests owing to its less-restrictive requirements [76]. 

## 4. Results

Based on the diffusion and interaction phenomena summarized in Section 2, the sensor temperature emerges as one of the most important parameters, if not the primary one, affecting the response and recovery times. Temperature increases are beneficial both for faster adsorption/recovery kinetics and for larger diffusion coefficients. The temperature dependence of the response and recovery times extrapolated from the analyzed dataset is reported in Figure 3. Since the measured response and recovery times depend on the characteristics of the measurement method, colors and symbols are used to highlight the adopted setup for each paper/data-point. 

Boxplots summarizing the statistical distribution of response and recovery times within each class of the measurement method descriptor are also shown in Figure 3. 

Looking at the vertical axes of the plots reported in Figure 3, the difference between t_RES_ and t_REC_ immediately emerges, the former being shorter than the latter. This is in good agreement with the kinetics of interaction processes discussed in Section 2. It is worth noting that, despite the well-established effect of the sensor temperature on response/recovery transients, a clear relationship between T_S_ and t_RES_ or t_REC_ does not emerge. As shown in Appendix B, Figure A1, this is also true for other quantitative parameters, namely, C_EtOH_ and r_p_, which are often recognized in individual papers as key factors controlling the t_RES_ and t_REC_ [48,77,78] and for the size of the sensor chamber and the gas flow, which are clearly related to the setup transients (Figure A2). This may not be surprising. Owing to the large differences between different papers regarding the adopted experimental setups, measurement parameters and material properties, the effects of individual parameters overlap one another and become hard to recognize. Considering that these arguments are also likely to apply to the measurement method, it is remarkable how clearly this experimental parameter affects t_REC_, as shown in the boxplot of Figure 3d. The flow-through method results are systematically slower than those of the static one, the two methods being characterized by median values of 125 and 28 s, respectively. Moreover, the first quartile of the dynamic method is 103 s, which is larger than the 3rd quartile of the static method (44 s), further indicating the occurrence of a meaningful difference between the two distributions. Before proceeding further, it is worth noting that the data present outliers and that data distributions are not symmetric, i.e., they are characterized by not-negligible Skewness values, and the medians differ from the respective means. These features are typically considered to be indicators that the considered distributions are not Gaussian, and statistical tools not assuming this kind of distribution should be preferred for the analysis. In this context, medians and quartiles are considered more reliable than means and standard deviations as estimators of the central tendency and spread of data distribution. The Mood’s median test is a suitable tool for comparison [76]. These arguments apply to the data shown in Figure 3 and, in general, to all of the data treated in the present work.

Concerning t_RES_ (Figure 3a,b), the median extrapolated for the dynamic setup is larger than the one of the static setup—30 vs. 12 s—but the difference between the two groups of data is not as marked as it was for t_REC_. The further analysis through the Mood’s median test returns a *p*-value of about 0.037 under the null hypothesis of not-distinguishable medians. Though this is smaller than 0.05, the threshold usually adopted to reject a null hypothesis, the distance from the threshold is not quite marked, and prudence should be adopted before assessing the occurrence of an effect. However, considering the evidence that emerged for t_REC_, it seems reasonable to suppose that the same setup dependence also holds for t_RES_, despite the differences being less marked owing to the smaller values.

Given the pronounced effects of the measurement method on the sensor recovery times, we further investigated the dependency of t_REC_ on other parameters separately for dynamic and static measurement configurations. In particular, we carried out this analysis focusing on morphological and heating method effects. 

Figure 4 reports the boxplots resuming the distribution of t_REC_ for different heating methods. At first, it is worth nothing that the different heating methods have not been evenly used with the dynamic and static setups. For example, the furnace has been used only in combination with the dynamic setup and not with the static one. On the other hand, the embedded coil as a heater has been widely used in combination with a static setup and has very rarely been used with the dynamic one. Moreover, it is interesting to observe that the meander-boxplot in Figure 4a is composed of two sub-clusters, one characterized by t_REC_ > 100 s and the other by t_REC_ < 100 s, which correspond to the meander-boxplots for the dynamic (Figure 4b) and static (Figure 4c) setups, respectively. These arguments have important implications for comparing the heating method effects on t_REC_.

Concerning the dynamic setup, the distributions shown in Figure 4b suggest that the use of a furnace may provide faster recovery times with respect to the use of an embedded meander as a heater. This may be reasonably explained in terms of the diffusion phenomena introduced in Section 2. Indeed, in a furnace setup, the whole environment (furnace) is warmed at the temperature of the sensor device, and thermo-diffusion is not expected to occur owing to the absence of the temperature gradient around the sensor device. 

Unfortunately, in static conditions (Figure 4c), it is not possible to investigate whether the use of a furnace has the same effects owing to the lack of a statistic for this type of heating method. Nonetheless, it can be observed that sensor layouts employing a coil and a meander as embedded heaters show similar medians (the *p*-value of the Mood’s median test is around 0.5). Regarding the diffusion processes, these are expected to occur in both types of sensor layouts.

The morphological effects are shown in Figure 5. As was already observed for the heating method, the distributions of t_REC_ for different morphologies also show a dependence on the gas delivery system. Indeed, the nanowire boxplot shown in Figure 5a is composed of two subgroups, one characterized by t_REC_ > 100 s and the other by t_REC_ < 100 s. With the exception of a few outliers, these two groups correspond to nanowires measured with a dynamic and a static setup, respectively. Moreover, in Figure 5a, the nanoparticle boxplot is characterized by a very large number of outliers, which strongly decreases the splitting of the dataset into those related to the static and dynamic setups. 

Concerning the dynamic setup (Figure 5b), the t_REC_ values extrapolated for nanowire appear as statistically smaller than those extrapolated for nanoparticles. The difference is not as marked as the difference between the furnace and the embedded meander for the heater category. Nonetheless, the Mood’s median test returns a *p*-value of about 0.009, which lets us reasonably speculate that morphological effects emerge not only at the level of individual papers but also in this general comparison between different papers. The open morphology featured by nanowire networks offers a very open structure, with voids that are much larger than those typically encountered in nanoparticle-based layers, hence leaving room for an efficient diffusion through the whole volume of the sensing layer. 

In the case of static setups (Figure 5c), the boxplots do not show any clear difference between the nanowires and nanoparticles (*p*-value ≈ 0.8). This may arise from the lower values recorded with the static setups compared to those recorded with the dynamic ones. Indeed, in the former case, most of the t_REC_ values span between 10 and 100 s, while in the latter case, the majority of data exceed 100 s, and a few are also above 1000 s. The low t_REC_ values recorded for the static setups approach the values of t_RES_, which were too low to extrapolate a statistically meaningful difference among the classes of the considered variables. In addition to the crystallite shape, their size is another fundamental parameter in gas sensing. Decreasing the crystallite size is often reported as fundamental to improving the response intensity [79,80]. They also speed up the diffusion of oxygen ions through the bulk of individual nanostructures to restore the bulk-surface equilibrium once this has been altered by exposure to the target gas (Section 2.5). On the other hand, in some cases, small crystallites are indicated as worsening the response and recovery times. This is because smaller nanostructures often imply smaller voids between interconnected nanostructures and hence a less efficient diffusion [12,53].

The effects of the crystallite size on t_REC_ have been investigated in conjunction with the setup and crystallite morphology effect. Figure 6a shows that, if we look at the whole set of data, a clear relationship between t_REC_ and d_c_ does not emerge. The situation is different if we look at the data grouped by the adopted experimental setup. Concerning static setups, no clear trend emerges, but this may be due to the low t_REC_ values of this type of setup, as was already observed in Figure 6 for morphological effects. Interestingly, for dynamic setups, a tendency of t_REC_ to decrease as d_c_ increases emerges. This tendency has been further investigated in conjunction with crystallite shape effects. Figure 6b shows that decoupling the two effects is not trivial. If we consider all the morphologies, a trend emerges, but the trend is less marked if we look at the nanowire and nanoparticle morphologies separately. Indeed, in statistical sense, the investigated nanowires have a diameter larger than that of the nanoparticles, and it is hard to assess whether the recovery time reduction is due to the size or shape effect or a combination of both. This difficulty may partially arise from the fact that both approaches, i.e., the use of nanostructures with larger diameters and the use of nanowires instead of nanoparticles, are beneficial for diffusion phenomena [11,12,53].

## 5. Discussion

The results reported in Section 4 show that the comparison of the results published in different papers is as important as it is difficult. 

Difficulties arise because it is hard to decouple the individual roles played by the phenomena described in Section 2. Among these effects, only thermo-diffusion and the gas–MOX interaction feature an asymmetric behavior between the response and recovery times, but these two effects are not sufficient to explain the observed results.

Concerning the recovery times, the measurement method emerges as a fundamental parameter affecting t_REC_. Considering the lack of systematic information about the setups used in the literature, it is hard to provide a solid explanation; however, the role of fluid dynamic conditions may be reasonably speculated upon. Dynamic setups are designed to keep the fluid dynamic conditions constant; the balance of different mass flow controllers connected to different certified bottles is changed to properly control the atmospheric composition, but their sum is kept constant. In this way, the sensor response is effectively related to the change in atmosphere composition, with no or minimal influence from the flow conditions. The situation is different in static setups, in which the injection or removal of the target analyte occurs through a break of the static condition, and it is thus likely to impact the fluid dynamic conditions as well and, in turn, impact the diffusion phenomena discussed in Section 2. Especially for the recovery, some papers report that the test chamber is deliberately opened, thus introducing a macroscopic perturbation of the fluid-dynamics.

Diffusion, particularly thermo-diffusion, and crystallite morphology produced statistically appreciable effects on t_REC_ in measurements carried out using the dynamic method. While the present analysis did not reveal any clear statistical evidence for other parameters and t_RES_, this does not mean that these parameters are not meaningful. Indeed, the present results are not in contrast with results reported in individual papers about the related sensors/materials. It simply means that these effects are coupled with effects induced by many other parameters and are shadowed by the large experimental differences occurring among different papers. Only in a few cases, as detailed in Section 4, is it possible to obtain global, statistically meaningful evidence over the whole set of considered papers. 

In general, the present results show that, to compare results from different papers, t_REC_ and t_RES_ should be considered not as related to the given sensor or material but to the whole experimental setup. An analysis considering only one or a few of the effects discussed in Section 2 may not be complete. Though specifically designed setups have been developed to obtain evidence of some of the mentioned phenomena, papers dedicated to the development of innovative materials are typically carried out with commercial [73,81,82] or home-made instruments [83,84], which do not have such a specificity. In this sense, a better knowledge about the used setups would help in better framing the results in a more general context. Of course, the first step is an accurate and detailed report of the experimental facilities and parameters. In order to better understand and account for all the flow, diffusion and gas–MOX interaction phenomena underlying the macroscopic t_RES_ and t_REC_, finite element simulations represent a very useful tool for modeling the whole system being tested (measurement chamber, sensor and material). Simulations encompassing the transients of the setup and of the gas–MOX interaction have been recently carried out [85]. The authors focused on a specific experiment involving a static measurement setup, a porous Fe_2_O_3_ layer deposited on a tubular substrate provided with an embedded coil as a heater [86]. The model included the gas flow through the chamber, the diffusion through the porous MOX and the interaction with the MOX surface for both the oxygen and the target gas (acetone), combining their reactions and reaction-dependent concentrations at the MOX surface [85]. It may work as a reference and a stimulus for additional works in the field, which are addressed to study other setups and/or include other effects in the simulations such as thermo-diffusion.

## 6. Conclusions

This paper reports a statistical analysis of the literature results published in the period of 2015–2020 about the response and recovery times recorded for ethanol chemiresistors based on pure SnO_2_. 

The results can be schematically summarized as follows:Stating whether a given sensor/material exhibits faster or slower response/recovery times with respect to the literature is not as simple as it may seem. Irregular data distributions, including several outliers and/or sub-clusters, are observed owing to the broad range of experimental differences that may occur between the works and their impact on sensor transients;Concerning the recovery times:
the experimental setup, whether of the static or dynamic type, emerges as the main factor influencing the response and recovery transients, with the static setups being statistically faster than the dynamic ones;by splitting data into the two setup categories, the distributions become more regular, and effects related to the crystallite morphology and the heating method emerge and can be reasonably explained in terms of diffusion phenomena;
Concerning the response times:
t_RES_ values are smaller than those of t_REC_, and this makes the differences between data distributions less pronounced with respect to the t_REC_ case;despite difficulties induced by the small values, it seems reasonable to suppose that the setup effects observed for t_REC_ also work for t_RES_;


In addition to these results, the present statistics may provide a useful reference for benchmarking purposes, both for the specific case analyzed here (ethanol sensing with pure-SnO_2_ chemiresistors) and in general for gas sensors, for which SnO_2_ is often used as a reference material.

## Figures and Tables

**Figure 1 sensors-22-06346-f001:**
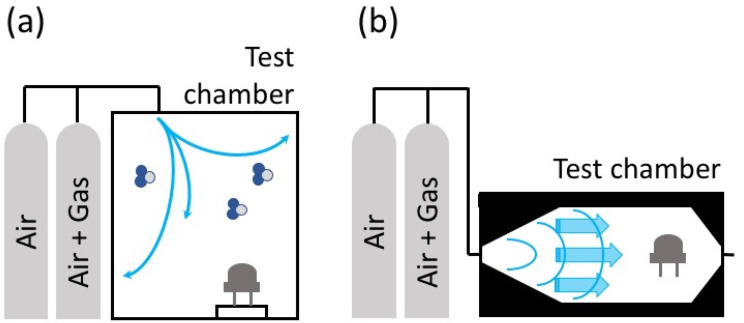
Schematic representation of diffusion and flow phenomena occurring in test chambers. (**a**) flow through pipelines and diffusion inside the test chamber volume; (**b**) flow through pipelines and the test chamber volume.

**Figure 2 sensors-22-06346-f002:**
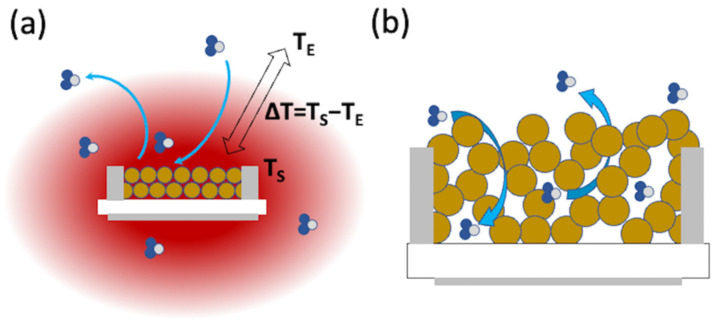
Schematic representation of diffusion phenomena occurring in proximity and through the metal oxide layer. (**a**) Diffusion through the temperature gradient surrounding sensors with an embedded heater owing to the difference between the sensor temperature (T_S_) and the environment temperature (T_E_); (**b**) diffusion through the pores of the metal oxide layer owing to the different gas concentrations in the environment and inside the sensing layer.

**Figure 3 sensors-22-06346-f003:**
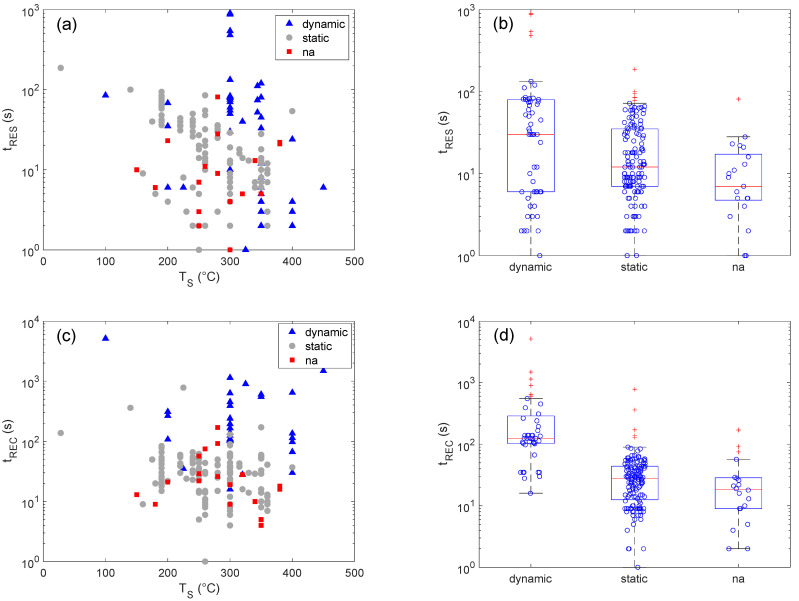
Sensor temperature and measurement setup effects on the response and recovery times. Plot of response time values vs sensor temperature (**a**) and related boxplot (**b**). Plot of recovery time values vs sensor temperature (**c**) and related boxplot (**d**). In (**a**,**b**), colors indicate the different measurement setups, and ‘na’ stands for ‘not available’—for those papers that did not provide enough detail to properly classify the measurement method.

**Figure 4 sensors-22-06346-f004:**
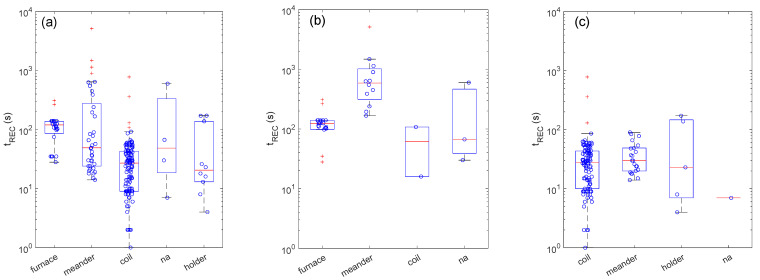
Heating method effects on recovery times (t_REC_) for measurements. Boxplots showing the distribution of t_REC_ for different heating methods for the whole set of measurements (**a**) and for the sub-set of measurements carried out with dynamic (**b**) and static (**c**) setups. In all plots, ‘na’ stands for ‘not available’—for those papers that did not provide enough detail to properly identify the class of the descriptive parameter.

**Figure 5 sensors-22-06346-f005:**
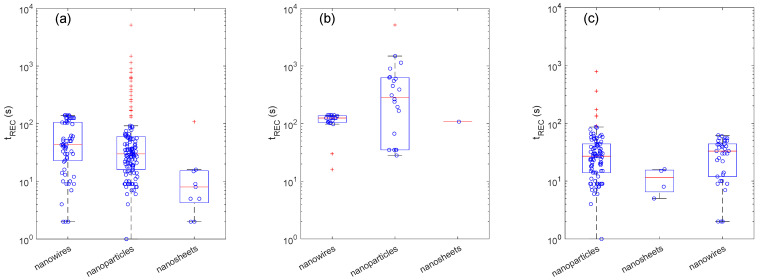
Effects of crystallite morphology on recovery times (t_REC_). Boxplots showing the distribution of t_REC_ for different crystallite morphologies for the whole set of measurements (**a**) and for the sub-set of measurements carried out with dynamic (**b**) and static (**c**) setups.

**Figure 6 sensors-22-06346-f006:**
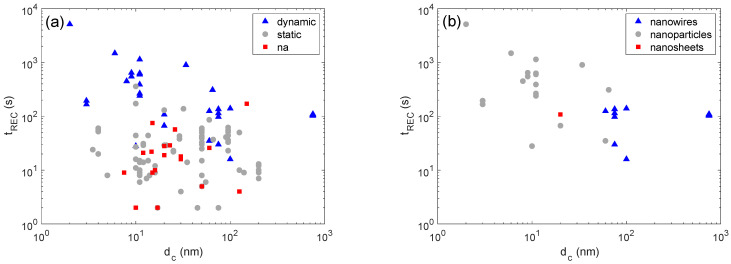
Effects of crystallite size (d_c_) on recovery times (t_REC_). Plot showing the dependence of t_REC_ on dc for the whole set of measurements; different colors identify different setups (**a**). Plot showing the dependence of t_REC_ on d_c_ for the set of measurements carried out with dynamic setups; different colors identify different crystallite shapes (**b**). In all plots, ‘na’ stands for ‘not available’—for those papers that did not provide enough detail to properly identify the class of the descriptive parameter.

## Data Availability

The list of papers used as source for statistical analysis is reported as Appendix A.

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
