# Peer review of "A Statistical Analysis of Response and Recovery Times: The Case of Ethanol Chemiresistors Based on Pure SnO2"

_sensors, 2022, doi:10.3390/s22176346_

Round 1

Reviewer 1 Report

In this work, the author aim at providing a general view of the response and recovery times of SnO2 gas sensors for ethanol sensing by analysing the results published in the period from 2015 to 2020. Due to the wide variety of experimental setups, both for the gas injection and the heating system, and of the sensor morphologies, the author uses statistical analysis to extract global trends in the understanding of the times. The work is of great interest as it clearly demonstrates the strong influence of the experimental conditions on the obtained response and recovery time, which mostly hides the physical response and recovery times of the sensor. Several points should however be addressed to improve the completeness and impact of the work.

1) In the diffusion models presented in section 2, the authors do not introduce the diffusion of oxygen species as interstitial defects into the metal oxide nanostructure. This model has been proposed to explain long response times, e.g. in [Hernandez-Ramirez et al., https://doi.org/10.1002/adfm.200701191]. This model should be added here for completeness.

2) The author do not provide any numerical values for the time scales expected for the various mechanisms described in section 2.

3) When describing the numerical variables that could influence the response and recovery time in section 3, the authors should add the size of the chamber. Given that the flow/diffusion through the chamber seems to be the most important factor limiting the recovery time, and given the strong influence of the chamber size on the expected response time (see e.g. the simulations of Ref. 18), this parameter should analysed.

In addition, here are minor comments to be addressed:

a) Line 133: Check again the validity ranges of the molecular diffusion vs. Knudsen diffusion, as the current ranges in the manuscript (0.1 < Lmfp/2rp < 1 and Lmfp/2rp >= 0.1) overlap. Moreover, molecular diffusion should correspond to larger value of rp as compared to Knudsen diffusion.

b) Line 136: For the evaluation of Lmfp (50 - 150 nm), provide the values of the used pressure and molecule size.

c) Line 247 "the sensor temperature emerges as one of the most important parameters": The effect of the temperature is clearly indicated for the adsorption/desorption mechanism (section 2.4), but it is not mentioned at all for the diffusion mechanisms. The author should add the expected temperature dependence of the various diffusion mechanisms in sections 2.2 and 2.3.

d) Lines 256-258, "in Figure 1, the difference between tRES and tREC immediately emerges, the former being appreciable shorter than the latter. This is in good agreement with the kinetics of thermally activated process discussed in Section 2.": The asymmetry between tRES and tREC is not explicitely mention in section 2. The author should explain why the "kinetics of thermally activated process" is expected to lead to an asymmetry. This is not obvious from the diffusion, which is expected to be symmetric, while the the adsorption/desorption mechanism could be asymmetric, but predicting which time is larger is far from obvious.

Reviewer 2 Report

In this work, the authors use a statistical approach to screen and compare the effects of nano-microstructure, gas concentration, the optimum working temperature and measurement methods for gas sensors’ response/recovery times. The work is focused on chemiresistor devices via referring to pristine SnO2 detect ethanol, which is the most studied material among metal oxides and offers a suitable statistic. There are some issues should be addressed.
1. The reference source in the line 99 and 108 should be corrected.
2. The response/recovery times are also related to the electron transfer rate in the sensing materials. Whether the voltage provided by the different instruments also matters.
3. The application of electrode sheets of different materials leads to the transfer efficiency of current in the sensors, whether this factor affects the response/recovery times of sensors should be discussed.
4. The surface reaction rate is related to the concentration of surface adsorption-reaction sites, which is greatly influenced by the sensing microtopographies. The authors should give a reasonable discussion.
5. The oxygen molecules and ethanol molecules are simultaneously introduced during dynamic test process, while oxygen molecules are infused before ethanol molecules are infused during the static testing. How it affects the response/recovery times? 
